# Genetic Architecture of Grain Yield-Related Traits in Sorghum and Maize

**DOI:** 10.3390/ijms23052405

**Published:** 2022-02-22

**Authors:** Wodajo Baye, Qi Xie, Peng Xie

**Affiliations:** 1State Key Laboratory of Plant Genomics, Institute of Genetics and Developmental Biology, The Innovative Academy of Seed Design, Chinese Academy of Sciences, Beijing 100101, China; bayekobo@gmail.com; 2University of Chinese Academy of Sciences, Beijing 100049, China

**Keywords:** QTLs, grain yield, sorghum, maize, marker-assisted breeding

## Abstract

Grain size, grain number per panicle, and grain weight are crucial determinants of yield-related traits in cereals. Understanding the genetic basis of grain yield-related traits has been the main research object and nodal in crop science. Sorghum and maize, as very close C4 crops with high photosynthetic rates, stress tolerance and large biomass characteristics, are extensively used to produce food, feed, and biofuels worldwide. In this review, we comprehensively summarize a large number of quantitative trait loci (QTLs) associated with grain yield in sorghum and maize. We placed great emphasis on discussing 22 fine-mapped QTLs and 30 functionally characterized genes, which greatly hinders our deep understanding at the molecular mechanism level. This review provides a general overview of the comprehensive findings on grain yield QTLs and discusses the emerging trend in molecular marker-assisted breeding with these QTLs.

## 1. Introduction

Increasing world population growth, which is expected to reach 9 billion, will require increasing food production to be doubled by 2050 [1,2]. Facing limited cultivated land, a lack of water resources and extreme environmental deterioration, humans have been seeking sustainable ways to maximize crop production to feed the soaring population [3]. Since world food production is mainly expected from crops, one way to meet this challenge is to improve the unit yield of crops [4]. Sorghum (*Sorghum bicolor* L. Moench), as the fifth largest food crop, guarantees food security for ~5 million people living in drought, saline-alkali, and barren areas worldwide [5,6]. Maize (*Zea mays* L.), as one of the three largest crops, feeds one-third of the world’s population [7]. Currently, sorghum and maize have also become available for feed, fodder, and emerging bioenergy [8,9]. Both sorghum and maize are the closest and relatively recent domesticated crops from their wild progenitor, which diverged from each other approximately 11.9 million years ago and distantly diverged approximately 50 Mya from their rice ancestor [10].

Grain yield is one of the most complex quantitative features with a varied genetic basis and is affected by multiple genetic components and environmental factors [11]. Grain weight is a grain yield component with high heritability. Grain size (grain length, grain width, and grain thickness) also significantly contributes to grain yield [12,13,14]. Grain yield is also determined by grain number per panicle in sorghum, maize, and other cereals [15,16,17,18]. In addition, the grain-filling rate and multiple-ovary development at anthesis also affect grain weight and grain number per panicle, respectively [19].

The complex genetic architecture of grain yield-related traits is often controlled by multiple genes known as quantitative trait loci (QTLs), which are genetic loci responsible for phenotypic variation of a quantitative trait. QTLs generally refers to the genomic position of genes that control quantitative traits, which are located by flanking genetic markers. Researchers found one or more QTLs next to genetic markers by looking for the relationship between genetic markers and quantitative traits of interest. In other words, QTLs and flanking markers should be linked. QTL mapping refers to locating QTLs on the genetic map by using several methods, such as interval mapping (IM) and composite interval mapping (CIM). The Kosambi function is used to determine the distance between QTLs and genetic markers. A QTL with LOD value more than 2.5 usually represents a credibly genetic locus for a trait. WinQTLCart (version 2.5) is one of the most used linkage mapping software [20] and the results obtained are more relatively accurate when compared to others.

Map-based cloning and genome-wide association studies (GWASs) are two effective ways to dissect complex quantitative traits and identify candidate loci across the whole genome [21,22]. The discovery of major QTLs for grain yield-related traits is a key goal of crop genetic research. The fast development of DNA markers and the sorghum and maize genome sequencing projects make QTL detection for grain yield traits easier. In particular, in the last two decades, numerous studies have used a genetic linkage mapping method to map QTLs associated with grain yield-related traits in sorghum and maize [20,23]. However, cloned genes related to sorghum and maize yield-related traits are not well studied, maybe due to complex polygenic control and variable environmental effects. This review summarizes current knowledge of initial and fine-mapped QTLs and functionally characterized genes associated with sorghum and maize grain yield-related traits to provide a more in-depth understanding of genetic networks in crop floret development and grain yield improvement.

## 2. Increased Grain Yield through Crop Domestication

Crop domestication is a complex combination of various selection pressures, with frequent fluctuations in introgression levels throughout human history [24]. Archaeological and genetic studies suggested that sorghum [25] and maize [26] were domesticated in different regions from their wild ancestors. Sorghum originated from four prominent wild sorghum (*Sorghum bicolor* ssp. *verticilliflorum* (L.) Moench) races, notably *aethiopicum*, *verticilliflorum*, *virgatum,* and *arundinaceum* [27], in Ethiopia and Sudan 8000 years ago [25] and then swiftly disseminated to Southern Africa and Asia. Maize was domesticated 9000 years ago in southwest Mexico from the wild progenitor teosinte (*Zea mays* ssp. *Parviglumis*) [26,28]. According to recent comparative genomics techniques, maize and sorghum have tight genetic collinearity and distantly share a common ancestor, which was closely related to 11.9 MYA [10,29]. They also hypothesized that both share a highly common identity between duplicated loci of maize and the corresponding orthologous region of sorghum [10]. A current study also shows high homologous similarity associated with seed protein quality based on phylogenetic analysis of 511 single-copy homologs between sorghum kafirins and maize zeins [29,30].

During crop domestication, symbolic phenotypic modifications, such as increased grain size and grain weight, were key targets by creating genetic variety from wild species [31]. Wild species generally have small grains to avoid being found and eaten by animals to ensure their own transmission and reproduction [32]. However, they were gradually domesticated into large and heavy seeds to meet the demand for human food resources [33]. Genetic changes in other grain yield-related traits have also accumulated to feed the world. A case in point is that grain number per panicle of both sorghum and maize cultivars increased more than their wild progenitors [34,35,36]. Bommert et al. reported that grain number per panicle was negatively correlated with grain size and some other inflorescence architectures [37]. It is difficult to greatly improve current sorghum and maize production due to the bottleneck effect of a few favorable alleles [36,38].

Nodal positioning, simple pathways, minimal pleiotropic effects, and selection on standing genetic variations from wild species are main factors for promoting convergent molecular domestication in crops [39]. Convergent increased grain yield in crop evolution is generally caused by genetic mutations within orthologous genes. For example, rice *GS3*, located at a key nodal position of the G-protein pathway, was first cloned to be a major gene for grain length and grain weight [40,41]. Then, *ZmGS3* [42] in maize and *SbGS3* [43] in sorghum were also reported to be associated with grain weight in diverse varieties. Naturally, rare and excellent mutations originating from wild progenitors have accidently increased grain size, grain weight, and grain number. These beneficial alleles were consciously selected and broadly used in cultivar breeding by humans. For example, eight QTLs for kernel weight and four QTLs for kernel row number were identified through a segregated population derived from teosinte and maize [44]. In sorghum, nine QTLs have been detected for grain weight and grain width derived from wild *Sorghum Virgatum* and domesticated *Sorghum bicolor* [45]. In a recent study, fifteen significant QTLs for grain yield-related traits were identified from a population derived from wild *Sorghum halepense* and *Sorghum bicolor* [46].

## 3. Genetic Dissection of Grain Yield-Related Traits in Sorghum and Maize

### 3.1. Sorghum

Grain yield-related traits have been extensively investigated in sorghum over the last two decades, and a large number of QTLs for grain weight and grain size are now mapped. Sorghum grain weight has shown to be highly inversely related to grain number per panicle [47,48] and significantly correlated with grain length and grain width [43]. To date, over 168 primarily mapped QTLs associated with grain size have been reported in sorghum. Among them, 155 QTLs located on ten chromosomes have been mapped for thousand grain weight (TGW) in various independent studies published between 1995 and 2021 [13,34,43,45,46,47,49,50,51,52,53,54,55,56,57,58,59,60,61,62,63,64,65,66] (Figure 1a and Appendix A). Few QTLs related to grain size (21 QTLs for grain length, 23 QTLs for grain width, and 26 QTLs for grain number per panicle) have been identified. Most grain size QTLs contribute minor phenotypic effects [43,45] and are unevenly distributed among ten chromosomes (Figure 1b).

Because of the quantitative nature, sensitivity to environmental effects, and complex genetic networks of grain yield-related traits, inadequate works have fine-mapped these QTLs in sorghum (Table 1). To date, few reports have attempted to predict candidate genes associated with sorghum grain weight and grain size QTLs. *qTGW1a* was recently defined as a 33-kb region on the long arm of chromosome 1 by flanking markers *SM010165* and *SM010171*, which contain three candidate genes. One of them, *SORBI_3001G341700*, encodes an atypical G-protein γ subunit, and as a causative gene of *qTGW1a*, *it* negatively controls grain weight in sorghum. The wild-type *qTGW1a* allele showed decreased grain size, plant height and grain yield in transgenic overexpression rice lines [43]. This locus is homologous to rice *GS3* and maize *ZmGS3*, which have also been shown to be negative regulators of grain length and grain weight [41,42,67,68]. Tao et al. [69] also confirmed a major QTL for grain size, which harbors the *SbGS3* gene in a diverse sorghum panel. Two previously identified QTLs, *QKwt.uga-C* [65] and *qtl1bGW* [51], were also reported to be colocalized with *qTGW1a* [43]. *GS3* can explain up to ~70% of the phenotypic variation in grain length among 180 rice varieties, while sorghum *qTGW1a* and maize *ZmGS3* can explain 4–10% and 8% of the phenotypic variation in grain weight, respectively [42,43]. These results suggest that both sorghum and maize Gγ-like proteins cause minor effects on grain yield despite rice Gγ-like proteins.

The other major QTL for sorghum grain weight, *qGW1*, was fine mapped into a 101 kb targeted region on the short arm of chromosome 1 flanking by markers *SB00037* and *SB00219. qGW1* could explain 20–40% of phenotypic variations across multiple genetic backgrounds and various environments. However, among the 13 putative candidate genes in this region, none of them had homologs to previously reported grain weight-related genes in other plants [13]. Even so, we compared reported genes that were domesticated in sorghum evolution [90] and found one possible gene (*Sobic.001G038900*) with a strong selection signal located in the fine-mapped region. This hypothesized gene encodes a DUF567 protein with high expression at the anthesis stage, and it should be further confirmed by researchers. Moreover, a novel QTL within a 5 Mb genomic region on chromosome 5 for TGW was recently discovered by Boyles et al. [50]. A closer look revealed that the highest-ranking marker was linked with a remorin protein (*SORBI _3005G188400*), which was primarily expressed in early young panicle and developing embryo tissues [91].

### 3.2. Maize

Kernel weight is confirmed to be significantly correlated with kernel length in maize [92,93,94]. Kernel size (same as sorghum grain size) also positively contributes to the end-use quality of maize [95]. Compared to sorghum, much more improvements have been made to identify major QTLs or genes for grain yield-related traits in maize. Currently, over 1920 QTLs associated with kernel size and kernel weight reported from 1994 to 2021 have been identified [11,12,44,53,70,71,72,73,83,92,93,94,96,97,98,99,100,101,102,103,104,105,106,107,108,109,110,111,112,113,114,115,116,117,118,119,120,121,122,123,124,125,126,127,128,129,130,131]. Among these loci, 528 QTLs related to hundred kernel weight (HKW), 299 QTLs associated with kernel length, 386 QTLs associated with kernel width, 250 QTLs related to kernel thickness, 118 QTLs associated with kernel number per row (KNPR), and 329 QTLs related to kernel row number (KRN) were detected (Figure 1c and Appendix A). All these QTLs were unevenly distributed among the ten chromosomes of maize (Figure 1d).

A total of 20 loci for grain yield-related traits were fine mapped in maize (Table 1). *GW4.05*, *qhkw5-3*, *qGW1.05,* and *qKW9* are four major QTLs for kernel weight that have mapped on chromosomes 4, 5, 1, and 9, respectively. *qGW4.05* mapped into a 279.6 kb interval between markers *ND16* and *ND19* that explained 23.9% of phenotypic variation and contained two annotated genes. *GRMZM2G039934* is predicted to be a candidate for *qGW4.05*, which encodes a putative leucine-rich repeat receptor-like protein kinase. Six polymorphic sites in the CDS were significantly associated with kernel weight and size between parental lines (HZS and LV28). *qGW4.05* exhibits the best pleiotropic effect on kernel weight, length, and width [86]. The other major QTL for kernel weight, *qhkw5-3*, was recently fine mapped into a 125.3 kb physical region between markers *InYM20* and *InYM36*. Within this locus, six genes have been annotated [87]. *qGW1.05* was narrowed down to a 1.11 Mb target region flanking the SSR markers *umcl601* and *umcl754*, which explains 20.94% of the phenotypic variation in KW and contains 30 predicted genes [88]. *qKW9* was recently narrowed down to a 20 kb target region between markers *M3484* and *M3506*, with 3 annotated genes. One of them, *Zm00001d048451*, which encodes a PLS-DYW-type PPR protein, is the causal gene for kernel weight [89].

*qKL1.07*, *qKL9,* and *qKL-2* are three major QTLs for kernel length. *qKL1.07* on chromosome 1 was delimited into a 1.6 Mb genomic fragment flanking markers *ML194* and *ML162*, which harbors 5 genes. *qKL1.07*, *GRMZM2G348452 (ZmCKX10)*, encodes a cytokinin oxidase that explains 11% of phenotypic variations in kernel length [81]. *qKL9* is the other major QTL on chromosome 9, which was identified to increase kernel length and HKW [82]. Liu et al. [94] previously reported that *qKL9* was highly associated with kernel length in maize in multiple environments by using F_2:3_ families. This was repeatedly confirmed by mapping the BC_2_F_2_ and BC_3_F_1_ populations with 16.09% phenotypic variance in kernel length. This locus was eventually delimited into a 942 kb region between markers *C9-54* and *C9-58*, which contained 24 annotated genes. Among them, *Zm00001d046723*, which encodes an expansin-A20 protein, is a potential candidate. Three indel variations in the 5′UTR of this gene induce significant differences in expression levels between near-isogenic lines (NILs) Mc^qKL9−A^ and Mc, which are also associated with kernel length [82]. Another recent QTL, *qKL-2*, was physically mapped to a 1.95 Mb interval between markers *mk3106* and *mk3114* on chromosome 9 containing 40 genes. Only one gene, GRMZM2G006080, is supposed to be a putative candidate for *qKL-2*, which encodes the receptor-like protein kinase FERONIA [83]. The receptor kinase FERONIA has also demonstrated itself as a signaling pathway that negatively regulates the elongation of integument cells and then controls seed size in *A. thaliana* [132].

Two major QTLs, *qKW7* and *qKW9.2* for kernel width, were fine mapped on chromosomes 7 and 9, respectively. A major QTL, *qKW7*, was narrowed down to a 647 kb target region between markers *7H-16* and *7F-5*, which contains 4 annotated genes. Among these kinases, *GRMZM2G114706* encodes an ankyrin protein kinase that positively regulates kernel weight and kernel width in maize [73]. A major locus *qKW7* for kernel width was recently fine-mapped using a series of backcross populations derived from a cross between YE478 and Huangzaosi. It was divided into two tightly linked intervals (*qKW7a* and *qKW7b*) with opposite phenotypic effects. *qKW7a* had a minor additive effect and was highly influenced by the environment. However, *qKW7b* had a high LOD value with additive effects and a larger kernel width due to harboring the YE478 allele, which is a promising locus for kernel width. The locus was eventually narrowed down to a 59 kb region, which contains a single gene. *Zm00001d020460* encodes a putative zinc finger homeodomain (ZF-HD) transcription factor whose gene expression level is associated with kernel development [84]. *qKW-9.2* for kernel width was mapped into a 630 kb region between markers *FSR6* and *MSR36* on chromosome 9, which explained 5.23–11.26% of phenotypic variances in 4 different locations [72].

*KNE4*, *qKNR6*, *qKNPR6,* and *qKN* are four major QTLs for KNPR that have been fine mapped on chromosomes 4, 6, and 10. *KNE4* was localized to a 440 kb genomic region flanking *umc1667* and *umc2135*. Only *Zm00001d052399*, which encodes a long-chain acyl-CoA synthetase and explains 26% of phenotypic variation in KNPR, is supposed to be a candidate gene [85]. *qKNR6* was recently mapped into a 110 kb target region flanking markers *M6* and *M8*, which harbors two candidate genes. Among these genes, *Zm00001d036602* is a causative gene for *qKNR6*, which encodes a serine/threonine protein kinase that regulates KNPR variation at different gene expression levels [74]. *qKNPR6* was narrowed into a 198 kb target region on chromosome 6 between markers *N6M19* and *umc1257*, which contains 6 candidate genes. *qKNPR6* explains 56.3% of KNPR variation and exhibits pleiotropic effects on ear length and kernel weight [75]. *qKN* was identified as a 480 kb region between markers *bnlg1360* and *umc1645* [76]. A favorable allele derived from parent 178 increased the KNPR by 6.08–10.76% [76].

The other major QTLs, *KRN1.4*, *krn1*, *qkrnw4*, *KRN4*, *qKRN5b,* and *qKRN8*, for KRN were fine-mapped on chromosomes 1, 4, 5, and 8, respectively. *KRN1.4*, a major locus that explained 50.48% of the phenotypic variance for KRN, was narrowed into a 203 kb region containing 7 predicted candidate genes. Among them, *Indeterminate spikelet1* (*ids1*) is a probable candidate gene that encodes an APETALA2 (AP2)-like transcription factor that regulates inflorescence branching, floral meristem determinacy, and spikelet meristem determinacy [77]. *qKRN5*, as a major QTL for KRN, was previously reported on chromosome 5. Currently, through fine mapping in some advanced backcross populations, *qKRN5* was dissected into two tightly linked loci, *qKRN5a* and *qKRN5b*. *qKRN5b* was a major QTL with a higher additive effect than *qKRN5a* and subsequently fine mapped into a 147.2 kb target region flanked by markers *SC3603d1* and *SC14631*, which harbored three putative candidate genes. Among them, *Zm00001d013603* encodes an endonuclease/exonuclease/phosphatase family protein that can hydrolyze phosphatidylinositol diphosphates *and* was identified as the causal gene of *qKRN5b* based on expression analysis and sequence variation between the two parental lines [78]. The other major QTL of *krn1* has been delimited into a 6.6 kb genomic region on chromosome 1 flanked by *SNP1* and *SNP2* markers. A single gene located in *krn1*, *Zm00001d03462*9, which encodes an (AP2) transcription factor, controls KRN in maize [79]. *KRN4* was fine-mapped into a 3 kb region between markers *M6* and *M8*. Within this region, a 1.2 kb transposon-containing insertion was present in the parental line H21^NX531^, which may function in increased KRN since this indel was strongly associated with KRN variation in diverse maize inbred lines [80]. *qkrnw4* was located at a 33 kb interval between the *Ch4.200-1* and *Ch4. K-2* target region on chromosome 4, which contains two possible candidate genes. *Zm00001d052910* encodes a putative protein and was supposed to be a candidate for KRN based on its gene expression and bioinformatics analysis [71]. Recently, *qKRN8*, a novel QTL for KRN in maize, was fine-mapped into a 520 kb target region and localized an interval between markers *umc2571* and *umc2593* on chromosome 8, which harbors six annotated genes. Among these genes, *Zm00001d010007* encodes a START domain-containing protein as a causal gene for *qKRN8* since the differential expression pattern was found in immature ears of NILs qKRN8^Lian87^ and qKRN8^V54^ [70].

## 4. Functionally Characterized Genes Associated with Grain Yield-Related Traits in Sorghum and Maize

In the last two decades, genome sequencing and DNA markers have led to rapid progress in cloning genes underlying grain yield-related traits. To date, a total of 30 genes have been cloned with grain yield-related traits in sorghum and maize (Table 2). Zou et al. [43] recently reported the only cloned gene, *qTGW1a*, for sorghum grain weight. *qTGW1a* (*SORBI_3001G341700*), the ortholog to rice *GS3* [40,67] and maize *ZmGS3* [42], encodes a G-protein γ subunit located at the N-terminus [43]. Overexpression of the sorghum *qTGW1a* gene was shown to be a negative regulator of grain weight in sorghum. A 5-bp insertion in the fifth exon of *qTGW1a* in the parental line LTR108 with a large grain weight results in a 61-amino-acid truncation in the C-terminal domain [43]. However, the authors showed that the natural truncated qTGW1a version caused heavier grain weight, which is inconsistent with the fact that rice truncated *GS3-4* caused shorter grain length. This should be further carefully confirmed by the authors since the C-terminal domain can inhibit the negative regulatory function of the N-terminal domain [133]. *GS3* is the first identified gene for determining grain size on chromosome 3 in rice, and it has been shown to explain up to 72% of phenotypic variance [40,67]. Mutation at the second exon changes a cysteine codon (TGC) to a stop codon (TGA) in the large rice grain varieties, while *ZmGS3* has 5 exons encoding a protein with 198 amino acids. *ZmGS3* encodes a putative transmembrane protein that has been demonstrated to play a function in maize kernel formation through methods that differ from rice *GS3* [42,67]. Furthermore, *qTGW1a* is a minor gene for variations in sorghum grain weight due to its 4–10% phenotypic explanation [43].

Multiseeded (MSD) genes (*MSD1*, *MSD2,* and *MSD3*) regulate a critical pathway involved in sorghum grain number per panicle [16,17,18]. Compared to wild-type BTx623, identified mutants with multiseeded panicles can produce up to over 200% of grain number per panicle. The *MSD1* gene, *SORBI_3007G021140 (SbTCP16)*, encodes a plant-specific teosinte branched/cycloidea/proliferating cell nuclear antigen-domain (TCP) transcription factor that is involved in jasmonic acid (JA) biosynthesis. Even though the *msd1* mutant has a smaller grain weight, it compensates by producing more grains per panicle than the WT [156]. In addition, *SORBI_3001G121600 (SbTCP2)*, a class II TCP-domain protein of the CYC/B1 family and homolog to maize TB1 and barley VRS5/HvTB1, also affects grain number in sorghum [157,158]. *MSD2* (*SORBI_3004G078600*), an ortholog of the maize *tassel seed 2* (*TS2*) gene, encodes a lipoxygenase (LOX), which can catalyze the conversion of free α-linolenic acid (18:3) to 13(S)-hydroperoxylinolenic acid, which acts as the first committed step of JA biosynthesis [18]. JAs play an important role in regulating inflorescence development of plants, such as pollen development in Arabidopsis [159], embryo development in tomatos [160,161], and spikelet formation in rice [162]. Both *tassel seed 1 (TS1)-* and *TS2-*mediated sex determination in maize were identified [163,164,165]. *Ts1* encodes a plastid-targeted 13-lipooxygenase that catalyzes the first committed step in JA biosynthesis. In addition, *SbTs1* (*Ts1* ortholog of sorghum) is also highly correlated with the MSD phenotype in sorghum, which suggests a similar JA-mediated MSD pathway in crops [165]. Currently, new findings have reported that a six-row barley mutant *vrs2* shows an MSD-like feature by *Hv36160* and encodes the SHORT INTERNODES (SHI) transcriptional regulator, which is regulated by cross impacts of auxin, gibberellin and cytokinin [166]. Furthermore, *MSD1* can regulate the gene expression of *MSD2* by directly binding to its promoter [18]. Because JA biosynthesis is blocked in the *msd1* and *msd2* mutants, the pedicellate spikelet continues to develop into viable grains with complete spikelet fertility in both sessile spikelets (SSs) and pedicellate spikelets (PSs). Recently, a report showed that PS, as a functional organ, contributes to seed weight by translocating its photosynthetic products to the SSs in 10029 sorghums [167]. Restoring the fertility of PS ultimately causes a twofold increase in grain number per panicle through lost functions of *MSD1* and *MSD2* genes. [16,18]. *MSD3*, *SORBI_3001G407600*, which encodes a plastidial Ꙍ-3 fatty acid desaturase that acid desaturase enhances grain number by lowering JA levels. It catalyzes the conversion of linoleic acid (18:2) to linolenic acid (18:3), emphasizing the relevance of the JA regulatory module(s) in the control of the fertility of PS in sorghum. Loss-of-function mutations in *MSD3* possess dramatically reduced linolenic acid (18:3), which lowers the level of endogenous JA [17]. The *msd1*, *msd2*, and *msd3* mutant phenotypes can be recovered to sterile spikelets as the wild type by using methyl-JA treatment [16,17,18].

Although many QTLs for maize grain yield-related traits have already been identified, there have been few efforts to characterize the causative genes. However, several genes that regulate maize kernel development have been identified and functionally characterized. For example, *Emp16*, *Emp10*, *Emp11*, *Emp12*, *Dek2*, *Dek35*, *Dek37,* and *Zmsmk9*, which encode P-type pentatricopeptide repeat (PPR) proteins, are involved in the alternative splicing of mitochondrial genes to regulate maize kernel development. Loss of function of these genes can damage mitochondrial function and cause delayed kernel development, resulting in undersized kernels and decreased kernel yield [142,144,145,146,147,148,150,152]. The PLS-type pentatricopeptide repeat (PPR) genes *Emp4*, *Emp5*, *Emp7*, *Emp9*, *Dek36*, *Dek10*, *Dek39*, *PPR2263,* and *Smk1* encode proteins that function in the C-to-U editing of transcripts in mitochondria and chloroplasts [138,139,140,141,143,149,151,153,154]. Loss of function of these distinguished genes potentially delays embryo and endosperm development, leading to insufficient kernel yield [168].

*ZmSWEET4c*, *ZmNPF7.9*, *ZmCEP1*, *ZmBAM1d*, *qKW9*, *FEA2*, *KNR6*, *ids1/Ts6,* and *UB3* have also been responsible for grain yield-related traits in different manners. *ZmSWEET4c*, an ortholog of rice *OsSWEET4*, encodes a hexose transporter protein. It functions in sugar transport across the basal endosperm transfer cell layer (BETL), resulting in an increase in seed filling. Loss of function of both *ZmSWEET4c* and *OsSWEET4* is defective in seed filling, including a dramatic loss of endosperm. The locus also shows a strong selection signature in *ZmSWEET4c* during maize domestication [136]. Recent reports show that the nitrate transporter *ZmNPF7.9*/*NRT1.5* is involved in maize kernel development [169]. *ZmNPF7.9* is highly expressed in the cells of BETL. Loss of function of *ZmNPF7.9* results in faulty seed growth and aberrant starch deposition, resulting in a significant reduction in grain weight [135]. *ZmCEP1*, an ortholog of *OsCEP6.1*, as a short C-terminal encoded peptide (CEP), functions as a negative regulator of kernel weight in maize [134]. Compared to the wild type, overexpression of *ZmCEP1* significantly reduces plant height, kernel size and 100-kernel weight, while knockout of *ZmCEP1* by CRISPR/Cas9 noticeably enhances kernel yield-related traits [170]. The authors also showed that *ZmCEP1* participates in nitrogen metabolism, nitrate, sugar transport and auxin response pathways to affect kernel development [134].

*CLV1/BAM* family genes were reported to control shoot meristem development [171], floret number and fruit size in plants [172,173]. Currently, a major QTL, *qHKW1*, which encodes a CLAVATA1 (CLV1)/BARELY ANY MERISTEM (BAM)-related receptor kinase, accounts for 18.4% of phenotypic variation in maize HKW. Sequence variation revealed a positive selection signal with low DNA methylation in the promoter region of *ZmBAM1d*, which enhances its gene expression and increases HKW during maize evolution [137]. Furthermore, overexpression of the retromer complex subunit *ZmVPS29* results in a slenderer kernel morphology with a lower KW, a greater KNPR, and a larger grain yield per plant (YPP). *ZmVPS29* also played a crucial role in improving maize yield during domestication [174]. Another major gene, *qKW9*, encodes a DYW-PPR protein and is essential for C-to-U editing ndhB, a subunit of the chloroplast NADH dehydrogenase-like complex. A 13 bp deletion in the coding sequence of *qKW9* reduced photosynthetic activity, resulting in less maternal photosynthetic availability during the grain-filling process and a lower kernel weight [89]. *FASCIATED EAR2 (FEA2)*, an ortholog of *Arabidopsis CLAVATA2* (*CLV2*), encodes a leucine-rich repeat (LRR) receptor-like protein and regulates KRN by transferring signals from *ZmCLV3* (the homolog to *Arabidopsis CLAVATA3*) peptide ligand to *WUSCHEL (WUS)* homeodomain transcription factor [175,176]. The loss function of *FEA2* has irregular kernel rows, increased KNPR, and a much higher KRN than the wild type. The *fea2*-null mutant also shows a lower number of total kernels due to massive proliferation and resource competition [155]. *ids1/Ts6* and *Zm00001d034629* are orthologs of the wheat *q* gene, which encodes an AP2 domain protein. Overexpression of *Ts6*/*ids1* has been demonstrated to produce more SPMs and higher KRN [79]. *Ts6*/*ids1* can affect the transcripts of the downstream *fea3*, *fea4*, and *ra3* genes involved in SPM development in maize [79]. Genes, such as kernel row number 4 (*KRN4*) [80], *FEA2* [155], and GRF-interacting factor 1 (*GIF1*) [177], also control the development of maize ears. Among them, KRN4 can enhance kernel yield by increasing KNPR. *KRN4*, a 3-kb indel fragment located 60 kb downstream of the SBP-box gene *Unbranched3* (*UB3*), is responsible for variations in KRN by regulating *UB3* transcripts [80]. *qKNR6*, *Zm00001d036602*, encodes a serine/threonine protein kinase. Overexpression of *KNR6* increases kernel yield, while the presence of transposable elements and long terminal repeat retrotransposons in the regulatory region of *KNR6* decreases kernel yield in maize. *KNR6* can phosphorylate Arf GTPase-activating protein (AGAP) to also affect ear length and kernel number [74].

## 5. Conclusions and Prospective

The past two decades have identified hundreds of QTLs for grain yield-related traits in sorghum and maize (Figure 1 and Appendix A). Most of the fine-mapped QTLs and functionally characterized genes related to grain weight, grain length, grain width, and grain number per panicle in sorghum and maize are shown in Table 1 and Table 2 and Appendix A. Grain yield-related traits are generally controlled by a few major and multiple minor loci due to its quantitative nature and environmental factors. We should focus more on QTLs that steadily occur in various environments to promote fast-track discovery of candidate genes and further gene functions for grain yield-related traits. Highly efficient map-based cloning mainly depends on the number of recombinants and marker density. Once a target QTL is narrowed into a small region, to clone the candidate genes accurately, it is better to select the recombinants with extremely recessive phenotypes. When the number of candidate genes are mapped to single digits, expression pattern analysis, genetic variation detection, and functional prediction of coding proteins between the two alleles are three available approaches to obtain the final candidate. Genetic scholars currently also use BSA-Seq, SNP chip, and GWAS strategies to obtain the major genes quickly and accurately by high-throughput genotyping technologies, such as the low-cost Specific locus amplified fragment sequencing (SLAF-seq) [178,179,180,181].

Grain weight is a key component of grain yield-related traits in sorghum and maize. It is mainly determined by genetic factors and is also affected by environmental stress, to a certain extent. The abiotic stress, such as drought, salt, alkali, and barren, could cause indirect yield loss by producing ionic stress, osmotic stress, and oxidative stress [182]. For example, compared to *Sorghum propinquum*, there are beneficial alleles of at least ten QTLs of *Sorghum bicolor* that could increase total aboveground biomass and grain yield under salt stress conditions [183]. When drought occurs in maize during the flowering stage, asynchronism occurs in the anthesis and silking interval (ASI), which leads to serious yield losses. Researchers found the genetic manipulation of *ZmEXPA4* in developing maize ears could significantly reduce the ASI under drought conditions [184]. In addition to these factors, Murry et al. studied the genetic potential of sugar content in grain and stems of sorghum and found that the increase of stem sugar restricted the increase of grain yield. Thus, we can also use the QTLs related to sugar content to design grain sorghum (low-sugar stem) and sweet sorghum (high-sugar stem) [185].

Overall, this review summarizes noteworthy contributions to update the genetic basis of grain yield-related traits in sorghum and maize. The functionally characterized genes related to maize kernel yield may provide possible causal candidates for numerous collinear QTLs of sorghum, but further investigation should be given. With a more robust knowledge of these QTLs and their gene functions, we believe that genetic researchers and breeders could make good use of these molecular markers and beneficial alleles and remove linked harmful mutations to raise crop production by marker-assist breeding system (MAS). Furthermore, CRISPR-based genome editing [186], a new kind of revolutionary “5G-plant” technology and a transgene (DNA)-free approach to develop genetically modified organism (GMO), can rapidly allow for an optimum combination of various grain yield-related traits for designing ideal crops by directly, artificially, and accurately editing of target sequence in the future.

## Figures and Tables

**Figure 1 ijms-23-02405-f001:**
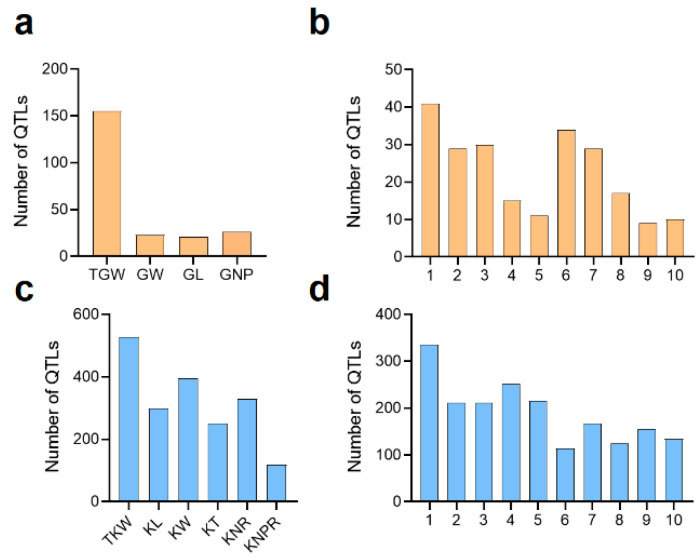
Number and distribution of quantitative trait loci (QTLs) for grain yield-related traits in sorghum and maize. (**a**) Number of QTLs associated with thousand-grain weight (TGW), grain width (GW), grain length (GL), and grain number per panicle (GNP) in sorghum. (**b**) Distribution of QTLs for grain size on ten sorghum chromosomes. (**c**) Number of QTLs associated with 1000-kernel weight (TKW), grain length (GL), grain width (GW), grain thickness (GT), kernel row number (KNR), and kernel number per row (KNPR) in maize. (**d**) Distribution of QTLs for grain yield-related traits on ten maize chromosomes.

**Table 1 ijms-23-02405-t001:** Fine-mapped QTLs associated with grain yield-related traits in sorghum and maize.

Crop	Trait ^a^	QTL	Chr ^b^	Marker Interval	Distance ^c^	Candi ^d^	Reference
Sorghum	GW	*qGW1*	1	*SB00037–SB00219*	101 kb	13	[13]
	GW	*qTGW1a*	1	*SM010165–SM010171*	33 kb	3	[43]
Maize	KRN	*qKRN8*	8	*umc2571–umc2593*	520 kb	6	[70]
	KRN	*qkrnw4*	4	*Ch4.200–Ch4.K-2*	33 kb	2	[71]
	KW	*qKW-9.2*	9	*FSR6–MSR36*	630 kb	28	[72]
	KW	*qKW7*	7	*7H-16–7F-5*	647 kb	4	[73]
	KNPR	*qKNR6*	6	*M6–M8*	110 kb	2	[74]
	KNPR	*qKNPR6*	6	*N6M19–umc1257*	198 kb	6	[75]
	KNPR	*qKN*	10	*bnlg1360–umc1645*	480 kb	1	[76]
	KRN	*KRN1.4*	1	*umc1737–cic001*	203 kb	7	[77]
	KRN	*qKRN5b*	5	*umc1365–umc2512*	147.2 kb	3	[78]
	KRN	*krn1*	1	*SNP1–SNP2*	6.6 kb	1	[79]
	KRN	*KRN4*	4	*M6–M8*	3 kb	2	[80]
	KL	*qKL1.07*	1	*ML194–ML162*	1.6 Mb	1	[81]
	KL	*qKL9*	9	*C9-54–C9-58*	942 kb	24	[82]
	KL	*qKL-2*	9	*mk3106–mk3114*	1.95 Mb	40	[83]
	KW	*qKW7b*	7	*M115.8–M116.7*	59 kb	1	[84]
	KRN	*KNE4*	4	*umc1086–M5*	440 kb	14	[85]
	GW	*qGW4.05*	4	*ND16–ND19*	279.6 kb	2	[86]
	GW	*qhkw5-3*	5	*InYM20–InYM36*	125.3 kb	6	[87]
	GW	*qGW1.05*	1	*umc1601–umc1754*	1.11 Mb	30	[88]
	GW	*qKW9*	9	*M3484–M3506*	20 kb	3	[89]

^a^ GW: grain weight; KRNL: kernel row number; KW: kernel width; KL: kernel length; KNPR: kernel number per row. ^b^ Chromosome. ^c^ physical distance (kb or Mb). ^d^ Number of candidate genes.

**Table 2 ijms-23-02405-t002:** Functionally characterized genes associated with grain yield-related traits in sorghum and maize.

Crop	Gene	Trait ^a^	Annotation	Variations	Ref ^b^
Sorghum	*MSD1*	GNP	TCP-domain TF	Missense mutation in *msd1-1*/2	[16]
	*MSD2*	GNP	lipoxygenase (LOX)	Nonsense mutation in *msd2-1*, missense mutation in *msd2-2*, nonsense mutation in *msd2-3*	[18]
	*MSD3*	GNP	ω-3 fatty acid desaturase enzyme	in *msd3-2*, nonsense mutation in *msd3-3*, alternative splicing in *msd3-1* and *msd3-4*	[17]
	*qTGW1a*	GW	G-protein γ subunit	5 bp insertion, frame shift	[43]
Maize	*ZmCEP1*	KS	C-terminal encoded peptide	Two frameshift mutations (1 bp insertion, 1 bp deletion) in *zmcep1*	[134]
	*KNR6*	KNPR	Protein kinase	substitution mutations	[74]
	*ZmNPF7.9*	KS	Nitrate transporter	Single nucleotide mutation (G to A)	[135]
	*ids1/Ts6*	KRN	AP2-domain TF	5 kb indel	[79]
	*ZmSWEET4c*	SF	Hexose transporter	Insertion in *zmsweet4c*	[136]
	*ZmBAM1d*	KW	CLV1/BAM receptor kinase	Insertion in *zmbam1d*	[137]
	*qKW9*	KW	DYW-PPR protein	Deletion in *qkw9*	[89]
	*PPR2263*	KS	DYW-PPR protein	Insertion in *ppr2263*	[138]
	*Emp4*	KS	PLS- PPR proteins	Insertions in the *emp4*	[139]
	*Emp5*	KS	PLS- PPR proteins	1.4 kb insertion in *emp5*	[140]
	*Emp7*	KS	PLS- PPR proteins	Insertion in *emp7*	[141]
	*Emp10*	KS	P-type PPR protein	431 bp deletion in *emp10*	[142]
	*Emp9*	KS	P-type PPR protein	Insertion in *emp9*	[143]
	*Emp11*	KS	P-type PPR protein	Insertion in *emp11*	[144]
	*Emp12*	KS	PPR protein	Insertion in *emp12*	[145]
	*Emp16*	KS	P-type PPR protein	Insertion in *emp16*	[146]
	*Dek2*	KS	P-type PPR protein	Insertion in *dek2*	[147]
	*Zmsmk9*	KS	P-type PPR protein	Frameshift mutation in *zmsmk9*	[148]
	*Dek10*	KS	E-subgroup PPR protein	5 bp insertion in *dek10*	[149]
	*Dek35*	KS	P-type PPR protein	Insertion in *dek35*	[150]
	*Dek36*	KS	E+ subgroup PPR	Insertion in *dek36*	[151]
	*Dek37*	KS	P-type PPR protein	Insertion in *dek37*	[152]
	*Dek39*	KS	PLS-PPR protein	Nonsense mutation in *dek39*	[153]
	*Smk1*	KS	PPR-E class protein	Missense, insertions in *smk1*	[154]
	*FEA2*	KRN	Leucine-rich repeat (LRR) receptor-like protein	Expression differences in *fea2*	[155]
	*UB3*	KRN	SBP-box TF	2 kb transposon-containing indel	[80]

^a^ GNP: grain number per panicle; GW: grain weight; KS: kernel size; KNPR: kernel number per row; KRN: kernel row number; SF: seed filling; KW: kernel. ^b^ Reference. TF: transcription factor.

## Data Availability

Not applicable.

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
