# Peer review of "Genetic Architecture of Grain Yield-Related Traits in Sorghum and Maize"

_ijms, 2022, doi:10.3390/ijms23052405_

Round 1

Reviewer 1 Report

A good review on grain yield-related traits in sorghum and maize.

A few minor revisions are needed.

  1. Figure 1 (c) legend needs to add "in maize"
  2. Table 1 and Table 2 list the QTLs and genes in both sorghum and maize. It is difficult for readers unfamiliar with the subject to determine which are from sorghum and which are from maize. I would suggest breaking the tables or splitting the tables with section titles indicating whether they are from sorghum or maize.

Author Response

A good review on grain yield-related traits in sorghum and maize.

A few minor revisions are needed.

> We are appreciated for the positive evaluation for the manuscript.

1. Figure 1 (c) legend needs to add "in maize"

> We added “in maize” in the legend of Fig. 1c accordingly.

2. Table 1 and Table 2 list the QTLs and genes in both sorghum and maize. It is difficult for readers unfamiliar with the subject to determine which are from sorghum and which are from maize. I would suggest breaking the tables or splitting the tables with section titles indicating whether they are from sorghum or maize.

> Thanks for your suggestion. We added section titles “sorghum” and “maize” in the Table 1 and Table 2.

Reviewer 2 Report

This is an interesting study and the authors have collected a unique dataset using cutting edge methodology. The paper is well written and structured. The science structure and overall theme of the manuscript are sound and acceptable. In my opinion, the overall concept is interesting and practically important. The paper is well written and is a worthy contribution that will be of interest to the readers of the IJMS journal.

The authors provide a comprehensive view on a large number of quantitative trait loci (QTLs) associated with grain yield. This review summarizes noteworthy contributions to update the genetic basis of grain yield-related traits in sorghum and maize.

The manuscript is interesting, associated with actual plant science trends. The presentation of the paper in the form of graphs and tables is clear and the analyses are well described.

All results are comprehensible. In general, the paper is clear and well-written. The quality of the manuscript is overall good considering the methodological approach, reliability of the results, and the ability of authors to discuss all the data properly.

Some arguments need clearer and tighter presentation, more understandable for a large spectrum of plant biologists.

The paper brings new aspects and novelties. The discussion could be improved to make the manuscript easy to follow, and more attractive and critical.

Discussion is not complete and written in favour of the authors suggested model. However, strong pieces of evidence are completely absent in this study.  I would have expected a more critical discussion of the results.  I invite authors to add additional references

I would have expected a discussion about environmental stress factors effects on plant regulation mechanisms. Please, include the missing information (research gaps). What are the alternative solutions?

Read/ use papers: https://doi.org/10.3390/genes12060797; http://dx.doi.org/10.4236/as.2014.511100; https://doi.org/10.3389/fpls.2019.01582; https://doi.org/10.1186/s12870-019-2164-5; https://doi.org/10.1016/j.stress.2021.100024, https://doi.org/10.1016/ j.jcs.2018.11.012; 923-934 http://dx.doi.org/10.4236/as.2014.511100

MINOR changes are needed in order to have a high-quality manuscript.

Author Response

This is an interesting study and the authors have collected a unique dataset using cutting edge methodology. The paper is well written and structured. The science structure and overall theme of the manuscript are sound and acceptable. In my opinion, the overall concept is interesting and practically important. The paper is well written and is a worthy contribution that will be of interest to the readers of the IJMS journal.

The authors provide a comprehensive view on a large number of quantitative trait loci (QTLs) associated with grain yield. This review summarizes noteworthy contributions to update the genetic basis of grain yield-related traits in sorghum and maize.

The manuscript is interesting, associated with actual plant science trends. The presentation of the paper in the form of graphs and tables is clear and the analyses are well described.

All results are comprehensible. In general, the paper is clear and well-written. The quality of the manuscript is overall good considering the methodological approach, reliability of the results, and the ability of authors to discuss all the data properly.

Some arguments need clearer and tighter presentation, more understandable for a large spectrum of plant biologists.

The paper brings new aspects and novelties. The discussion could be improved to make the manuscript easy to follow, and more attractive and critical.

Discussion is not complete and written in favor of the authors suggested model. However, strong pieces of evidence are completely absent in this study.  I would have expected a more critical discussion of the results.  I invite authors to add additional references

I would have expected a discussion about environmental stress factors effects on plant regulation mechanisms. Please, include the missing information (research gaps). What are the alternative solutions?

Read/ use papers:

https://doi.org/10.3390/genes12060797;

http://dx.doi.org/10.4236/as.2014.511100; (RETRACTED)

https://doi.org/10.3389/fpls.2019.01582;

https://doi.org/10.1186/s12870-019-2164-5; https://doi.org/10.1016/j.stress.2021.100024, https://doi.org/10.1016/j.jcs.2018.11.012; 923-934 http://dx.doi.org/10.4236/as.2014.511100 (RETRACTED)

MINOR changes are needed in order to have a high-quality manuscript.

> We are appreciated for the positive evaluation for the manuscript and many thanks for your suggestions.

  We have added some critical discussion in the “Conclusion and Prospective” section, especially a paragraph about environmental stress factors effects on plant regulation mechanisms accordingly (Line 408-447). The given references were also appropriately added in the reference section (Line 1027-1045).

Reviewer 3 Report

The review provides interesting insight in genetic architecture of two important grains. Their identification and location will help other researchers to use this informator for their work.

Genetic material contains huge amount information about numerous traits. These related to yields are important. Each new review combining data with many other studies and databases is wanted allowing easier finding required data.

For that reason, I found this manuscript worth publishing. It is well organized, written with good English language and containing all required information.

I only noticed that some Latin names of species are not in italic and some not start with capital names (genus name).

I recommend this manuscript for publication.

Author Response

The review provides interesting insight in genetic architecture of two important grains. Their identification and location will help other researchers to use this informator for their work.

Genetic material contains huge amount information about numerous traits. These related to yields are important. Each new review combining data with many other studies and databases is wanted allowing easier finding required data.

For that reason, I found this manuscript worth publishing. It is well organized, written with good English language and containing all required information.

I only noticed that some Latin names of species are not in italic and some not start with capital names (genus name).

I recommend this manuscript for publication.

> We are appreciated for the positive evaluation for the manuscript. We have checked the format of all Latin names. Now they are in italic.

Reviewer 4 Report

The mini-review discusses the different quantitative trait loci associated with sorghum and corn seed size.
The review is written in good language, and the theses presented in it are logical and correct.
The summarized information will help geneticists and breeders develop more productive varieties in the future.
The study of quantitative trait loci is an important area of modern genetics, and many readers would like to learn more about it.
Therefore, I ask the author in the introduction to describe in more detail what quantitative trait loci are and the methods and programs that allow them to be studied.  

Author Response

The mini-review discusses the different quantitative trait loci associated with sorghum and corn seed size.
The review is written in good language, and the theses presented in it are logical and correct.
The summarized information will help geneticists and breeders develop more productive varieties in the future.
The study of quantitative trait loci is an important area of modern genetics, and many readers would like to learn more about it.
Therefore, I ask the author in the introduction to describe in more detail what quantitative trait loci are and the methods and programs that allow them to be studied.  

 > We are appreciated for the positive evaluation for the manuscript and many thanks for your suggestions.

  We have added some detailed describe of QTLs definition, common methods and programs in the Introduction section accordingly (Line 42-60).